# Lactoferrin-Derived Peptide Chimera Induces Caspase-Independent Cell Death in Multiple Myeloma

**DOI:** 10.3390/cells14030217

**Published:** 2025-02-03

**Authors:** Young-Saeng Jang, Shima Barati Dehkohneh, Jaewon Lim, Jaehui Kim, Donghwan Ahn, Sun Shim Choi, Seung Goo Kang

**Affiliations:** 1Institute of Bioscience and Biotechnology, College of Biomedical Science, Kangwon National University, Chuncheon 24341, Republic of Korea; 2Department of Bio Health Science, Changwon National University, Changwon 51140, Republic of Korea; 3Division of Biomedical Convergence, College of Biomedical Science, Kangwon National University, Chuncheon 24341, Republic of Korea

**Keywords:** apoptosis-inducing factor, lactoferrin, lactoferrin chimera, multiple myeloma

## Abstract

Lactoferrin-derived peptide chimera is a synthetic peptide that mimics the functional unit of lactoferrin with antibacterial activity. Although LF has anticancer effects, to the best of our knowledge, its effects on multiple myeloma have not yet been studied. We explored the potential of a lactoferrin-derived chimera for multiple myeloma treatment, a malignant clonal plasma cell bone marrow disease. The lactoferrin-derived chimera effectively inhibited MM1S, MM1R, and RPMI8226 multiple myeloma cell growth, and induced the early and late phases of apoptosis, but not in normal peripheral blood mononuclear cells. Furthermore, the lactoferrin-derived chimera modulates the relative expression of genes involved in survival, apoptosis, and mitochondrial dysfunction at the transcriptional level. Mitochondrial analysis revealed that lactoferrin-derived chimera triggered oxidative stress in multiple myeloma cells, leading to reactive oxygen species generation and a decline in mitochondrial membrane potential, resulting in mitochondrial dysfunction. Although lactoferrin-derived chimera did not cause caspase-dependent cell death, it induced nuclear translocation of apoptosis-inducing factor and endonuclease G, indicating the initiation of caspase-independent apoptosis. Overall, the lactoferrin-derived chimera induces caspase-independent programmed cell death in multiple myeloma cell lines by increasing the nuclear translocation of apoptosis-inducing factor/endonuclease G. Therefore, it has potential for multiple myeloma cancer therapies.

## 1. Introduction

Lactoferrin (LF), a versatile glycoprotein enriched in milk and granules of neutrophils, has multiple biological roles, such as immunomodulatory, iron transport, anti-inflammatory, antibacterial, antiviral, and antitumor properties [1,2]. Initially, LF was shown to have a strong bactericidal function, mainly mediated by two functional domains: lactoferricin and lactoferrampin located in the N1 domain of bovine LF [3]. Furthermore, a chimeric form (lactoferrin-derived chimera, LFch) containing a part of lactoferricin (17–30 a.a), and lactoferrampin (265–284 a.a) was created by fusion and surpassed individual peptides in bactericidal activity at low doses and short incubation times [4].

In cancer therapy, LF demonstrates antitumor effects against numerous types of cancer, including breast cancer, cervical cancer, and leukemia [5,6,7]. The anticancer properties of LF include arresting the cell cycle, damaging the cytoskeleton, inducing apoptosis, and reducing cell migration [8]. Although the observed harm is evident, the underlying mechanisms responsible for these consequences are yet to be fully understood. LF exerts its anticancer effects via two potential mechanisms. It may trigger signaling pathways that can have detrimental effects on cells by binding to proteoglycans, glycosaminoglycans, and sialic acid, which are more abundant in cancer cells than in heathy cells [9]. The other mechanism may involve the disruption of iron metabolism, which is critical for cancer cell survival [10], due to the ability of LF to bind to iron ions [2].

The enhanced interaction of LFch with negatively charged model membranes amplifies the biological activity of these antimicrobial peptides through chimerization, mimicking the native structural orientation of the molecule, which is superior to the native LF form [4]. Therefore, we hypothesize that LFch possesses a strong anticancer effect similar to that of LF and is potentially even superior as it triggers apoptosis in both HepG2 and Jurkat cell lines [11].

In this study, we aimed to explore the antitumor effects of LFch in human multiple myeloma cell lines and clarify the underlying molecular mechanisms. This investigation may provide a theoretical basis for potential future clinical applications in multiple myeloma treatment, which is the second most common hematologic malignancy in adults and the most refractory form of malignancy [12].

## 2. Materials and Methods

### 2.1. Resources

Comprehensive information on the reagents and resources used in this study is presented in Table 1.

### 2.2. Cell Culture

Human MM cell lines, MM1S, MM1R, and RPMI8226, were obtained from the American Type Culture Collection (ATCC, Manassas, VA, USA) and the Korean Cell Line Bank (KCLB, Seoul, Republic of Korea). MM cells were maintained in KCLB medium, which consisted of RPMI 1640 medium supplemented with 10% fetal bovine serum (FBS), 1% penicillin/streptomycin, HEPES 1 M, and 7.5% sodium bicarbonate at 37 °C in a humidified atmosphere of 5% CO_2_. Passaging was conducted every 3 days to ensure optimal cell conditions. Human Peripheral blood mononuclear cells (PBMCs) were obtained from Stemcell Technologies (Vancouver, BC, Canada) and cultured in KCLB medium. For activation of primary human B cells, human PBMCs were treated with 5 µg/mL of anti-human IgM for 48 h.

### 2.3. Cell Viability and Proliferation

The viability of MM cells was measured using a Cell Counting Kit-8 (CCK-8) assay (Dojindo, Kumamoto, Japan). Cells (MM1S, MM1R at 1 × 10^5^ cells/well; RPMI8226 at 5 × 10^4^ cells/well) were treated with LFch (3µM) for 48 h. Viability was assessed using spectrophotometry at 450 nm. Proliferation analysis was conducted by staining the cells with the Ki67 marker, and flow cytometry was used for the analysis.

### 2.4. Cell Apoptosis Analysis

MM cells or activated human PBMCs were treated with LFch (3 µM) for 36 h, and apoptosis was detected using an Annexin V/PI apoptosis kit (eBioscience, San Diego, CA, USA) according to the manufacturer’s instructions. Briefly, cells were incubated with Annexin V (1 mL in 20 mL buffer, 1:20) for 20 min at 4 °C and then with PI (1 mL in 40 mL buffer, 1:40) for 5 min in the dark at room temperature.

### 2.5. Immunoblot Analysis

Total protein extraction was performed on cells (MM1S, MM1R, and RPMI8226 at 1 × 10^6^ cells/well) treated with LFch (3 µM). Cells were washed with PBS and lysed in 1× RIPA buffer (Cell Signaling Technology) supplemented with 1× phosphatase inhibitor mixture and 1× protease inhibitor mixture (GenDepot, Barker, TX, USA) by incubation for 30 min on ice. Nuclear extracts were prepared using a Nuclear/Cytosol Fractionation Kit (BioVision, Mountain View, CA, USA) according to the manufacturer’s instructions. After protein quantification, 20 µg of protein was resolved on NaDodSO4–PAGE gels, transferred onto a polyvinylidene fluoride membrane, and incubated with primary and secondary antibodies. Membranes were scanned using a ChemiDoc imaging system.

### 2.6. Mitochondrial ROS Detection

Changes in mitochondrial ROS were detected using MitoSox Staining. MM cells were treated with LFch (3 µM for 2 h. After washing with PBS, cells were incubated with 1 µM MitoSox solution and analyzed using a Flow Cytometer.

### 2.7. Mitochondrial Membrane Potential

Disruption of mitochondrial membrane potential was indicated using TMRE Staining. MM cells were treated with LFch (3 µM) for 2 h. After washing with PBS, the cells were incubated with 200 nM TMRE solution and analyzed using a Flow Cytometer.

### 2.8. RNA Sequencing

MM cells were treated with LFch (3 µM) for 24 h. Total RNA was extracted using the RNAeasy Kit (Qiagen, Valencia, CA, USA) according to the manufacturer’s instructions. The RNA library preparation and sequencing reactions were conducted according to standard Illumina protocols by Azenta Life Science (Suzhou, China). Sequencing libraries were generated using the Next^®^ Ultra RNA Library Prep Kit for Illumina^®^ (NEB) following the manufacturer’s protocol.

### 2.9. Data Production and Differentially Expressed Gene Identification

Triplicate RNA-seq data were generated from each of the three cell lines, MM1S, MM1R, and RPMI8226, under both control and LF-treated (LFch) conditions. After completing quality checks on the raw sequence data using FastQC (ver. 0.11.9), adapter sequences were trimmed using Trimmomatic (ver. 0.39). The cleaned read sequences were aligned to the reference genome (GRCh38/hg38) using the STAR (ver. 2.7.10a), and HTseq (ver. 0.11.2) was used to count aligned sequencing reads. Differentially expressed genes (DEGs) were identified using normalized read count data and the DESeq2 package in R (ver. 1.38.3) by comparing the control and LFch groups for each of the three cell lines. The two thresholds used for DEG identification were |log2(fold change)| ≥ 0 and adjusted *p*-value ≤ 0.01.

### 2.10. Data Analysis

Gene Ontology (GO) analysis was performed using DAVID (https://david.ncifcrf.gov/home.jsp, accessed on 9 August 2023), with terms exhibiting a *p*-value ≤ 0.05 being selected. All other data analyses were performed using R package (ver. 4.2.3). Specifically, principal component analysis (PCA) was conducted using the ‘factoMineR’ package (ver. 2.9), and heatmaps were constructed with z-scores of the normalized read count data using the ‘pheatmap’ package (ver. 1.0.12). Venn diagram analysis was conducted with the ‘VennDiagram’ package (ver. 1.7.3). The ‘ggplot2′ (ver. 3.4.2) and ‘ggrepel’ (ver. 0.9.4) packages in R were used to visualize volcano, PCA, and GO term bar plots.

### 2.11. Confocal Microscopy

The MM cells were incubated with LFch-biot for 1 h, and fixed with 4% paraformaldehyde. The cells were blocked and permeabilized with DPBS containing 10% goat normal serum and 0.1% Triton X-100. Cells were cytospun, and the slides were incubated with SAV-APC (BioLegend, San Diego, CA, USA) and stained with DAPI (Abbkine, Georgia, USA) for nuclear staining. The relative distribution of the fluorochromes was visualized using a supersensitive high-resolution confocal laser scanning microscope LSM880 with Airyscan (Carl Zeiss, Oberkochen, Germany).

### 2.12. Statistical Analysis

Data were analyzed using GraphPad Prism 10 (GraphPad Software, San Diego, CA, USA). Statistical analyses were performed using GraphPad Prism 10. Comparisons were performed using two-tailed Student’s *t*-test or two-way ANOVA. A *p*-value of ≤ 0.05 was considered statistically significant.

### 2.13. Data Code and Availability

The GEO accession number for the RNA-sequencing data reported in this paper is GSE273287.

## 3. Results

### 3.1. LF Chimera Inhibit the Growth of MM Cell Lines

To identify the cytotoxicity of LF and its derivative peptide, LFch, on MM, MM cell lines (MM1S, MM1R, and RPMI8226) were cultured with LFch, and cell growth was determined by colorimetric assays with various concentrations of LFch. The proliferation of all evaluated cell lines was severely decreased by LFch at a concentration of 3 mM and above (Appendix A). Therefore, we proceeded with the next analysis using the same concentration of LFch. Unlike LF, LFch significantly inhibited the growth of MM1S and RPMI 8226 cells, which was also observed in the drug-resistant MM1R cells (Figure 1A). LFch also decreased Ki67 expression in these cell lines, suggesting that LFch suppresses the proliferation of MM cells (Figure 1B). To further characterize the cytotoxicity of LFch in MM cell lines, we examined the effect of LFch on MM cell apoptosis. The annexin V-propidium iodide (PI) double-positive population significantly increased in all three MM cell lines, indicating that LFch induces both early- and late-phase apoptosis, and is pivotal for triggering programmed cell death (PCD) in MM cell lines (Figure 1C). In contrast, LFch did not induce apoptosis in activated CD19+ B cells (Figure 1D). Furthermore, LFch had no evident cytotoxicity towards CD3+ T cells and CD3-CD19- cells including monocytes in normal PBMCs, indicating no toxicity concerns for normal immune cells. Collectively, these results suggest the potential specificity of LFch as a promising therapeutic candidate for MM.

### 3.2. Reactive Oxygen Species (ROS)-Induced Mitochondrial Dysfunction Is Involved in LFch-Induced MM Cell Apoptosis

To understand the mechanisms underlying the antitumor effects of LFch in MM, we used RNA sequencing to reveal the genetic landscape. MM cells treated with LFch (3 µM) were subjected to a comprehensive omics study that illuminated potential gene targets and signaling pathways modulated by LFch. RNA sequencing revealed that LFch treatment increased the transcriptional levels of genes related to ROS-induced mitochondrial dysfunction and other cellular responses to cellular stress and DNA damage. In contrast, LFch decreased the expression of cell-survival-associated genes in all three MM cell lines (Figure 2A–C and Appendix A). Of the 34 common genes identified in MM1S, MM1R, and RPMI8226, genes involving mitochondrial dysfunction and oxidative phosphorylation, represented by MT_ND2, MT_ATP6, MT_ND1, MT_ND3, and iron metabolism and oxidative stress (*ISCU, FTL,* and *HMOX1*) revealed the impact of LFch on cellular iron storage, biogenesis of iron–sulfur clusters, and cellular response to oxidative stress (Figure 2D).

### 3.3. Mitochondrial Dysfunction Plays a Crucial Role in Cell Death Induced by LFch

An examination of LFch-mediated transcriptional changes in MM cell lines revealed the potential of LFch to induce cell death in MM cells through mitochondrial dysfunction. Mitochondrial damage causes the release of several mitochondrial proteins, including ROS, which initiate a PCD cascade [13]. ROS can damage cellular components and create a domino effect that disrupts the mitochondrial membrane potential (ΔΨm) [14]. We investigated this complex process by examining the role of ROS in LFch-induced mitochondrial dysfunction. LFch treatment significantly increased ROS levels, which were recovered by the ROS scavenger N-acetylcysteine (NAC) (Figure 3A). Consequently, cytotoxicity of LFch in MM cell lines was rescued by NAC treatment, highlighting the ROS-dependent mitochondrial destructive effect of LFch (Figure 3B). Furthermore, LFch treatment resulted in an increase in the TMRE^lo^ fraction of MMs, which is indicative of a reduction in mitochondrial membrane potential (Figure 3C). Collectively, LFch has the ability to trigger mitochondrial malfunction and programmed cell death (PCD) in MM cell lines.

### 3.4. LFch Exerts Antitumor Activities via Caspase-Independent Apoptosis

Next, to elucidate the mechanism by which LFch affects MM cells, confocal microscopy was performed. We detected penetration of LFch into the cytosol of all MM cell lines (Figure 4A). This may be because of the general characteristics of LFch as a peptide involved in mitochondrial damage. In the study of PCD pathways, we focused on the role of caspase-3 in LFch-induced apoptosis of MM cells. Notably, Western blot analysis revealed the significant absence of caspase-3 expression after LFch treatment. In contrast, the positive control, bortezomib (BTZ), which can induce caspase-3-dependent PCD [15], showed increased expression (Figure 4B). These results strongly suggest that LFch exerts antitumor effects on MM cells through caspase-3-independent apoptotic pathways. These findings contribute to our understanding of the complex cellular interactions of LFch and further highlight it as a promising candidate for the development of new caspase-3-independent targeted therapies for the treatment of MM. After identifying ROS-induced mitochondrial dysfunction, we sought to elucidate the downstream effects of LFch-induced mitochondrial dysfunction. We investigated the cytoplasmic-to-nuclear translocation of apoptosis-inducing factor (AIF) and endonuclease G (EndoG), which are known to trigger caspase-independent death pathways. We investigated the nuclear translocation of AIF and EndoG using Western blot. In all three MM cell lines, LFch augmented the nuclear translocation of AIF and EndoG (Figure 4C). This relocation not only increased their movement, but also played a central role in initiating chromatin condensation and DNA fragmentation, which are hallmarks of phase-1 apoptosis. DNA cleavage orchestrated by AIF and EndoG nuclear translocation underscores the role of LFch in caspase-independent death. This mechanism further supports the argument that LFch is a potent inducer of apoptosis in MM cells, providing a unique and promising opportunity for therapeutic interventions. These findings help understand the complex interactions of LFch in the cell and shed light on the specific molecular events that drive its antitumor activity. Identification of AIF and EndoG as key players in this process will provide valuable information for the development of targeted therapies for MM.

## 4. Discussion

Although treatment approaches for MM have undergone major advancements in recent years, patients with MM frequently experience relapse after one or more treatment regimens or develop drug resistance, along with accompanying side effects and long-term complications. Hence, novel therapeutic approaches are required to address the constraints associated with current treatments [12,16]. This study elucidates the anticancer characteristics of LFch, which effectively suppresses the proliferation of human MM cell lines, including MM1S, MMIR, and RPMI8226, and triggers PCD, but has no impact on normal human B cells. Additionally, LFch has been shown to have a diverse array of molecular targets that regulate the growth and survival of MMs.

LFch was initially created to mimic the functional components of LF and to function as an antimicrobial peptide (AMPs) [4]. AMPs have attracted considerable interest as promising anticancer drugs because of their distinctive characteristics and mechanisms of action. AMPs possess the crucial characteristic of specifically targeting and disrupting cancer cell membranes, allowing them to function as anticancer drugs [17]. This selectivity is mostly ascribed to disparities in the lipid composition of cancer cell membranes, in contrast to those of normal cells [18]. For example, cancer cells frequently display negatively charged phosphatidylserine on their outer membranes. They can interact with the positively charged nature of many AMPs, causing damage to the cell membrane and resulting in cell death. This process is similar to microbial membrane damage by AMPs, demonstrating a dual function in both antimicrobial and anticancer properties. Furthermore, certain AMPs that contain positively charged and lipid-loving amino acids damage the cell membrane and enhance the passage of hydrophobic mitochondrial membranes, leading to the initiation of apoptosis [19]. Similarly to other AMPs that have anticancer effects, LFch also rapidly internalizes into malignant B cells, but not into normal activated B cells, within 30 min. MM is characterized by hypersialylated cell surface membranes [20], and LF preferentially binds to sialic acids [9]. The LF-derived peptide, LFch, may exhibit selective binding to malignant B cells rather than normal B cells, and subsequently penetrate cancer cells. However, the underlying mechanisms require further investigation.

We demonstrated that LFch induces PCD in MM cells without caspase activation. This process occurs by facilitating the translocation of AIF and EndoG from the cytoplasm to the nucleus. Gene profiling of MM cells affected by LFch also demonstrated a notable increase in mitochondrial dysfunction. Induction of apoptosis in tumor cells has been widely acknowledged as a crucial strategy for cancer treatment. Nevertheless, inherent resistance to PCD and swift recurrence of tumors following an initial positive reaction to chemotherapeutic drugs are regarded as the most substantial constraints of treatments that rely on inhibiting cell growth as their main mechanism [21]. Consequently, there is an increasing interest in caspase-independent cell death and the discovery of substances that function largely through the caspase-independent pathway [22]. Understanding this process offers valuable insights into how cells can undergo PCD without the involvement of caspase enzymes, thereby identifying prospective targets for therapeutic interventions in disorders where the regulation of PCD is disturbed. Caspase-independent apoptosis is a crucial process in PCD that depends on mitochondrial malfunction and the activity of pro-apoptotic proteins, such as AIF and EndoG [23,24]. AIF and EndoG typically reside within the intermembrane gap of mitochondria. After receiving a signal for PCD, they are released into the cytoplasm owing to the breakdown of the mitochondrial membrane, and then move to the nucleus. Within the nucleus, AIF attaches to DNA and triggers the compaction of chromatin as well as the fragmentation of DNA on a wide scale [23]. EndoG can directly cut DNA, resulting in significant DNA fragmentation [24,25]. Notably, findings not presented in this context indicate that LFch is capable of initiating cellular processes within the nucleus and triggering PCD. Nevertheless, it is essential to recognize that mitochondrial malfunction and the related cell death mechanism may be just one of the multiple mechanisms contributing to LFch-induced cytotoxicity.

Proteasome inhibitors (PIs) and immunomodulatory agents (IMiDs), approved for the treatment of MM, also directly elicit cytotoxicity through distinct modes of action compared to LFch. Therefore, it is conceivable that a combination effect exists with LFch. Upon investigation of this potential with bortezomib (BTZ), LFch did not exhibit a combination effect at the optimal dose of BTZ. Both PIs and IMiDs generate oxidative stress in MM cell lines such as LFch; therefore, we assumed that their combination would not yield a combination effect with LFch. Rather, the LFch function induces cell death through a caspase-independent mechanism, which we anticipate will serve as an alternative to PI- or IMiD-resistant multiple myeloma. It is worthwhile to investigate the potential cytotoxic effects of LFch on PI- or IMiD-resistant multiple myeloma cell lines. In addition, further research is required to evaluate the efficacy of LFch in treating MM utilizing an in vivo tumor model. These are the limitations of our current study.

Collectively, our findings demonstrate that LFch triggers apoptosis in MM cells, including drug-resistant MM1R cells, via a caspase-independent pathway. Upon further investigation of LFch sensitivity to PI- or IMiD-resistant MM cell lines including MM cells derived from refractory patients, LFch may emerge as a new and focused strategy for overcoming drug resistance and enhancing patient outcomes in MM.

## Figures and Tables

**Figure 1 cells-14-00217-f001:**
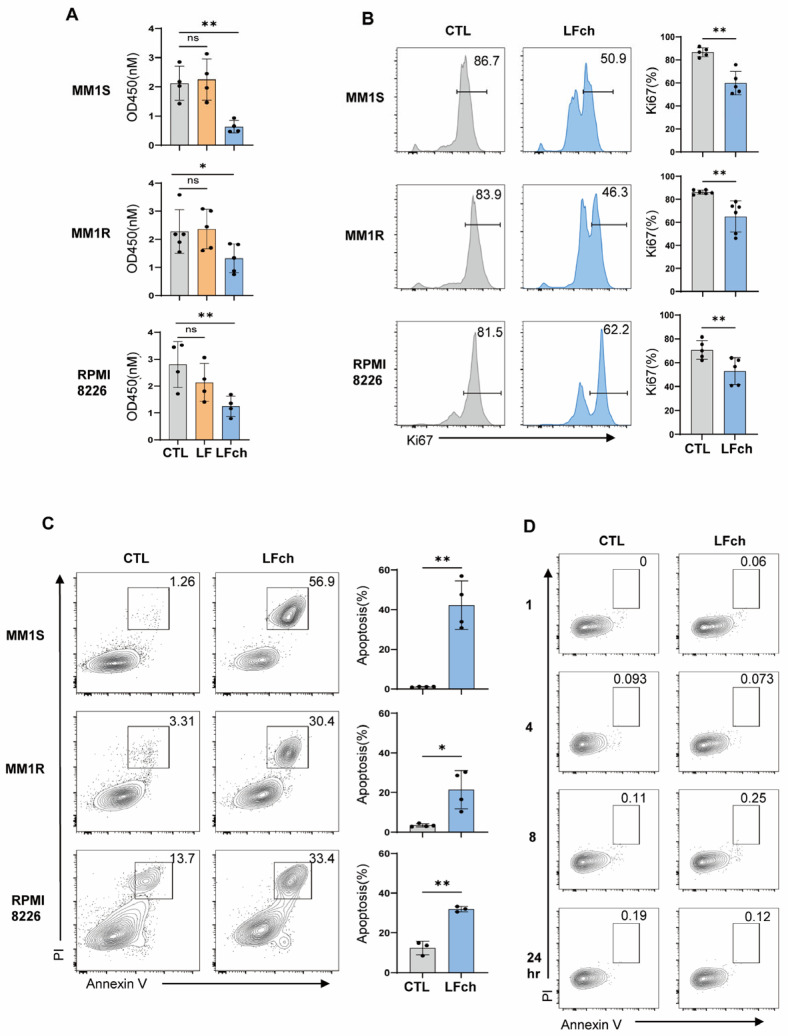
LFch induces cytotoxicity in MM cell lines, but not in normal PBMCs. MM cell lines (MM1S, MM1R, RPMI8226) were treated with 3µM of LF or LFch for 48 h. Growth inhibition by LF or LFch was assessed using the CCK-8 assay (**A**). Ki67 expression (**B**) and annexin V^+^PI^+^ apoptotic cells (**C**) were detected by flow cytometry. (**D**) Human PBMCs were cultured with 5 µg/mL of anti-human IgM for 48 h, then treated with LFch for indicated times and apoptosis was determined by flow cytometry. Data represent mean plus or minus the standard deviation (± SD) of 3 independent experiments. Statistical analyses were performed using the GraphPad Prism software (GraphPad Software, Inc., La Jolla, CA, USA). (* *p* < 0.05, ** *p* < 0.01, versus untreated control). LFch, lactoferrin-derived chimera; MM, multiple myeloma.

**Figure 2 cells-14-00217-f002:**
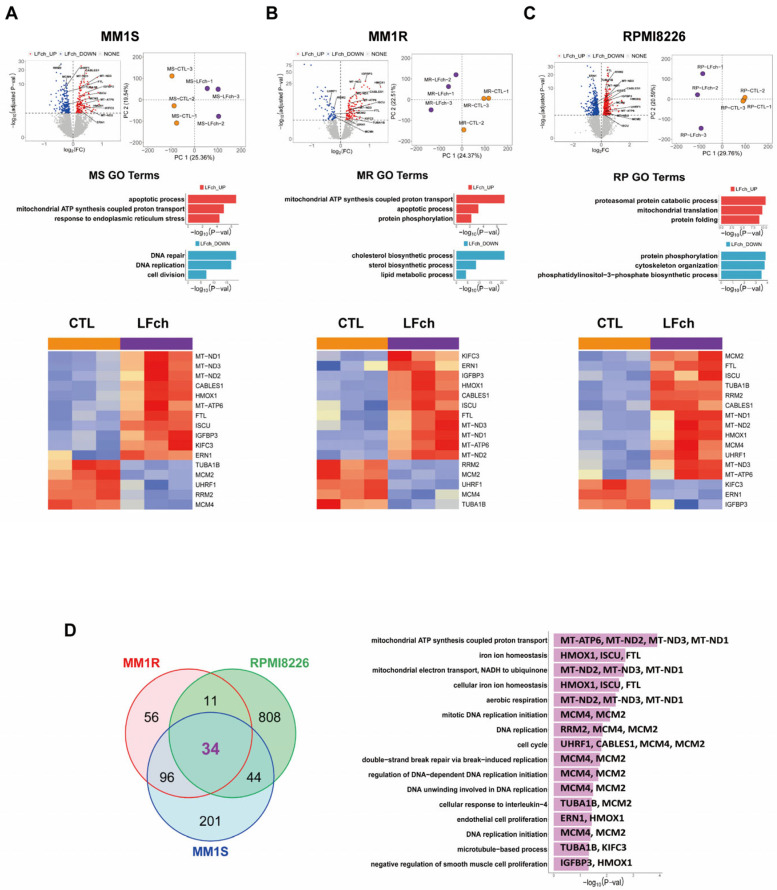
Transcriptomic analyses of LFch-affected MM cells. (**A**–**C**) RNA-sequencing results depict differentially expressed genes (DEGs) and Gene Ontology (GO) analysis after LFch (3 µM) treatment on three MM cell lines for 24 h. Volcano plot and heatmap of genes represent the distribution of DEGs compared to untreated control. Genes labeled in red are significantly upregulated, and blue-labeled genes are significantly downregulated. (**D**) The Venn diagram (left) shows the overlapping number of upregulated gene and 34 common DEGs in three multiple myeloma cell lines. The graph on the right represents the functional classification of 34 common DEGs. LFch, lactoferrin-derived chimera; MM, multiple myeloma.

**Figure 3 cells-14-00217-f003:**
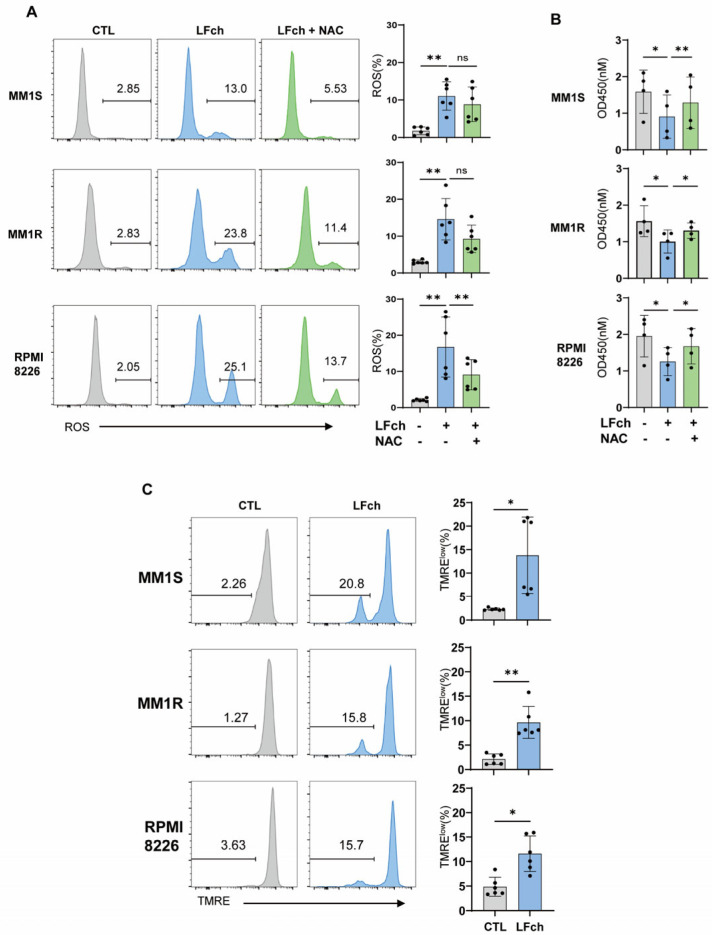
LFch increases the mitochondria ROS and membrane potential in MM cells. (**A**) MM cells were treated with LFch for 2 h and mitochondria ROS marked by MitoSOX probe was measured using flow cytometry. N-acetylcysteine (NAC, 10 uM) was added to the cultures for ROS scavenging. (**B**) MM cells were cultured as in A for 48 h, and cell viability was assessed by CCK-8 assay. (**C**) MM cells were treated with LFch for 2 h and quantitative analysis of mitochondrial membrane potential estimated by TMRE fluorescence. Data represent mean ± SD of three independent experiments (* *p* < 0.05, ** *p* < 0.01, versus untreated control). LFch, lactoferrin-derived chimera; MM, multiple myeloma.

**Figure 4 cells-14-00217-f004:**
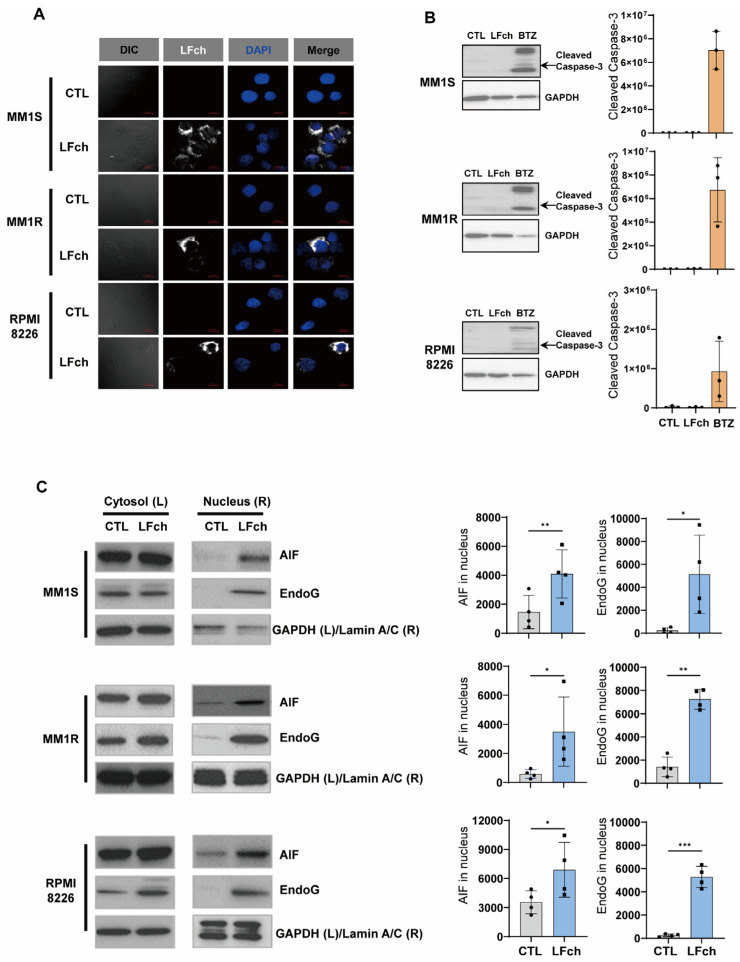
Induction of caspase-independent but AIF and EndoG-dependent apoptosis by LFch. (**A**) MM cells were treated with LFch for 30 min and cellular distribution of LFch was determined by immunocytochemistry. (**B**) The cleavage of caspase-3 was assessed by immunoblotting in MM cells treated with LFch for 24 h. Bortezomib (BTZ, 25 nM) served as a positive control. (**C**) The nuclear translocation of AIF and EndoG in cytosol or nuclear fraction of LFch-treated MM cells was determined by immunoblotting. Data represent mean ± SD of 3 independent experiments (* *p* < 0.05, ** *p* < 0.01, *** *p* < 0.001 versus untreated control). AIF, apoptosis-inducing factor; EndoG, endonuclease G; LFch, lactoferrin-derived chimera; MM, multiple myeloma.

**Table 1 cells-14-00217-t001:** The list of reagents and resources used for this investigation.

REAGENT or RESOURCE	SOURCE	IDENTIFIER
Antibodies		
Cleaved Caspase-3 (Asp175) (5A1E)	Cell Signaling Technology, Danvers, MA, USA	Ca# 9664
GAPDH	Proteintech, Rosemont, IL, USA	Ca# 60004-1
AIF	Cell Signaling Technology, Danvers, MA, USA	Ca# 4642
Endonuclease G	Cell Signaling Technology, Danvers, MA, USA	Ca# 4969
Lamin A/C	Cell Signaling Technology, Danvers, MA, USA	Ca# 2032
Ki67	BioLegend, San Diego, CA, USA	Cat# 151220
Annexin V	Invitrogen Life Technologies, Carlsbad, CA, USA	17-8007-74
PI	Invitrogen Life Technologies, Carlsbad, CA, USA	00-6990-42
Streptavidin-APC	BioLegend, San Diego, CA, USA	Cat# 405207
IgM	Southern Biotech, Birmingham, AL, USA	Cat# 2022-01
Chemicals, Peptides, and Recombinant Proteins		
Lactoferrin chimera (LFch) Sequence:FKCRRWQWRMKKLGK (RSKNKGFKEQAKSLLKWILD)-NH2	PEPTRON	No.# 23-47901
Lactoferrin	Morinaga Milk, Zama, Japan	GRN 464
Bortezomib	Sigma-Aldrich, St. Louis, MO, USA	504314
RPMI 1640 Medium	Gibco, Waltham, MA, USA	REF A10491-01
Fatal Bovine Serum	Gemini BioProducts, Woodland, CA, USA	Cat# 100-500
Penicillin-Streptomycin (10,000 U/mL)	Gibco, Waltham, MA, USA	Cat# 15140122
HEPES (1 M)	Gibco, Waltham, MA, USA	Cat# 15630080
Sodium bicarbonate	Sigma-Aldrich, St. Louis, MO, USA	Cat# S8761
RIPA Buffer (10×)	Cell Signaling Technology, Danvers, MA, USA	Cat# 9806
Phosphatase inhibitor mixture	GenDepot, Barker, TX, USA	
Protease inhibitor mixture	GenDepot, Barker, TX, USA	
DAPI	Abbkine, Atlanta, GA, USA	Cat# BMD0063
Critical Commercial Assays		
CCK8 solution	Dojindo Laboratories, Kumamoto, Japan	CK04
Foxp3/transcription factor fixation/permeabilization concentrate and diluent	eBioscience, San Diego, CA, USA	Cat# 00-5521-00
Annexin V/propidium iodide (PI) apoptosis kit	eBioscience, San Diego, CA, USA	Cat# 88-8007-74
MitoSOX™ Red Mitochondrial Superoxide Indicator, for live-cell imaging	Invitrogen Life Technologies, Carlsbad, CA, USA	Cat#M36008
Nuclear/Cytosol fractionation Kit	BioVision, Mountain View, CA, USA	Cat# K266-100
Bradford protein assay	Bio-Rad, Hercules, CA, USA	Cat#5000006
Foxp3 staining kit set	eBioscience, San Diego, CA, USA	Cat#00-5523-00
TMRE-Mitochondrial Membrane Potential Assay Kit	Abcam, Cambridge, MA, USA	Cat#ab113852
Cell Lines		
MM1S	ATCC, Manassas, VA, USA	CLR-2974
MM1R	ATCC, Manassas, VA, USA	CRL-2975
RPMI8226	KCLB, Seoul, Republic of Korea	KLCB No.10155
Software and Algorithms		
FlowJo v10.1	Tree Star, Ashland, MA, USA	
Prism v10	GraphPad Software, San Diego, CA, USA	
Chemi Doc imaging system	Bio-Rad, Hercules, CA, USA	
Image Lab	Bio-Rad, Hercules, CA, USA	

## Data Availability

The original data presented in the study are openly available in Gene Expression Omnibus (GEO) and accession number is GSE273287.

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
