# Peer review of "Lactoferrin-Derived Peptide Chimera Induces Caspase-Independent Cell Death in Multiple Myeloma"

_cells, 2025, doi:10.3390/cells14030217_

Round 1
Reviewer 1 Report
Comments and Suggestions for Authors
In this study, Jang et al. showed that lactoferrin-derived chimera (LFch), a chimeric form containing a part of lactoferricin (17—30 a.a) and lactoferrampin (265–284 a.a), induces caspase-independent programmed cell death in multiple myeloma (MM) cell lines by increasing the nuclear translocation of apoptosis-inducing factor/endonuclease G.
Although LF has anticancer effects, its effects on MM have not yet been studied. In addition, the authors hypothesized that LFch possesses a strong anticancer effect similar to that of LF and is potentially even superior as it triggers apoptosis in both HepG2 and Jurkat cell lines. Then, the authors determined the effects and suggest the potential for anti-MM drug. This reviewer thinks that the data presented include novel findings and would be beneficial for readers in community. This reviewer has the following suggestions that may help to improve the overall manuscript.
1. Figure 1D. The authors revealed that LFch induces cytotoxicity in MM cell lines, but not in normal PBMCs. However, the authors only showed the effect on CD19+ B-cells. The authors should determine the effect on other hematopoietic cells including T-cells, NK cells, and monocytes/macrophages.
2. Figure 4. The authors revealed that LFch induced reactive oxygen species (ROS)-mediated caspase-independent cell death. On the other hand, proteasome inhibitors (PIs) and immunomodulatory drugs (IMiDs), both of which are the key drugs for MM, can induce caspase-dependent apoptosis and trigger ROS production in MM cells (Sebastian et al. Blood,2017;129:991; Fink et al. Leukemia,2016;30:104). This reviewer thinks that LFch could enhanced the cytotoxicity in combination with them. The authors should determine the combined effects to MM cell lines or discuss with the possibility.
3. Discussion. The authors suggest that LFch has the potential to be a new and focused strategy for overcoming drug resistance (p. 14, lines 340-342). However, the authors only demonstrated the effect to MM.1R cells. This reviewer thinks that it is an overstatement.
Author Response
Please see the attachement

Reviewer 2 Report
Comments and Suggestions for Authors
The authors studied the effects of lactoferrin-derived peptide chimera (LFch) on myeloma cell lines showing increase of apoptosis in cell lines. Though interesting, there are some questions on this study
1. Is there dose effects and time effects of LFch on myeloma cell lines?
2. The authors used MM1S, MM1R, and RPMI8226 cell lines for the study, however, myeloma is a heterogenous disease with different pathomechanism and resistant strains. How about the effects of LFch on other cell lines?
3. Is there synergic effects of LFch with other anti-myeloma agents?
4. How about the effects of LFch on human myeloma samples?
Round 2
Reviewer 1 Report
Comments and Suggestions for Authors
-